# Effects of environmental conditions on healthcare worker wellbeing and quality of care: A qualitative study in Niger

Darcy M. Anderson [1]*, Ezechiel Mahamane[2], Valerie Bauza[1], Kairou Oudou Bilo Mahamadou[2], Lucy Tantum[1], Aaron Salzberg[1]

**1** The Water Institute at UNC, Gillings School of Global Public Health, The University of North Carolina at Chapel Hill, Chapel Hill, North Carolina, United States of America, **2** World Vision Niger, Nouveau Marche, Boulevard de la Liberté BP 12713, Niamey, Niger

* darcy.anderson@unc.edu

**Data Availability Statement:** Data are not publicly available for this manuscript to protect the privacy of research subjects. These are small healthcare

## Abstract

Environmental conditions (water, sanitation, hygiene, waste management, cleaning, energy, building design) are important for a safe and functional healthcare environment. Yet their full range of impacts are not well understood. In this study, we assessed the impact of environmental conditions on healthcare workers' wellbeing and quality of care, using qualitative interviews with 81 healthcare workers at 26 small healthcare facilities in rural Niger. We asked participants to report successes and challenges with environmental conditions and their impacts on wellbeing (physical, social, mental, and economic) and quality of care. We found that all environmental conditions contributed to healthcare workers' wellbeing and quality of care. The norm in facilities of our sample was poor environmental conditions, and thus participants primarily reported detrimental effects. We identified previously documented effects on physical health and safety from pathogen exposure, but also several novel effects on healthcare workers' mental and economic wellbeing and on efficiency, timeliness, and patient centeredness of care. Key wellbeing impacts included pathogen exposure for healthcare workers, stress from unsafe and chaotic working environments, staff dissatisfaction and retention challenges, out-of-pocket spending to avoid stockouts, and uncompensated labor. Key quality of care impacts included pathogen exposure for patients, healthcare worker time dedicated to non-medical tasks like water fetching (i.e., reduced efficiency), breakdowns and spoilage of equipment and supplies, and patient satisfaction with cleanliness and privacy. Inefficiency due to time lost and damaged supplies and equipment likely have substantial economic value and warrant greater consideration in research and policy making. Impacts on staff retention and care efficiency also have implications for health systems. We recommend that future research and decision making for policy and practice incorporate more holistic impact measures beyond just healthcare acquired infections and reconsider the substantial contribution that environmental conditions make to the safety of healthcare facilities and strength of health systems.

facilities in Niger with a limited number of healthcare workers. Given the limited number of health workers in rural Niger—plus rich contextual clues in the transcripts about their job duties, work environment, and local geography—making the data publicly available poses a risk that study participants may be identified. Furthermore, our consent process indicated that interview transcripts would be used exclusively by the study team for research purposes. We did not receive explicit consent to post the data to a public repository. Please use the following point of contact from the University of North Carolina at Chapel Hill's Institutional Review Board for questions or concerns regarding participant confidentiality and data availability: irb_questions@unc.edu.

**Funding:** This research was funded by World Vision through the Water Institute at UNC. World Vision provided support with designing the study sample, arranging travel and field logistics, and providing supervisory support to the data collection consultant. They had no role in the analysis or decision to publish. DMA is supported by a grant from the National Institute of Environmental Health Sciences (NIEHS) (T32ES007018). NIEHS had no role in the study design, data collection, analysis, or decision to publish.

**Competing interests:** The authors have declared that no competing interests exist.

## Introduction

Environmental conditions in healthcare facilities (HCFs) include water, sanitation, hygiene, waste management, energy, and other goods, services, and infrastructure that ensure facilities are safe, hygienic, and functional [1]. Safe environmental conditions protect patients, caregivers, healthcare workers, other staff (e.g., cleaners and waste handlers), and communities. The most well-recognized benefits of environmental conditions are preventing healthcare-acquired infections and slowing development of antimicrobial resistance [2–6]. While reductions of healthcare-acquired infections and antimicrobial resistance are important, they do not represent the full range of benefits. Environmental conditions are important for non-infectious outcomes, such as patient care seeking and healthcare worker safety and satisfaction [7–9]. Poor environmental conditions have economic consequences, as workers may pay out of pocket for supplies (e.g., personal protective equipment [PPE]) [10]. Environmental conditions also affect healthcare workers' ability to deliver care, such as electricity for lighting care at night or operating equipment [11].

While these non-infectious and non-health outcomes are important, there have been few attempts to comprehensively identify benefits beyond infectious outcomes. Furthermore, most research focuses on the patient perspective, with few studies on how environmental conditions affect healthcare workers or their ability to deliver quality care. More holistically understanding effects of environmental conditions is important for protecting healthcare workers and patients, prioritizing effective interventions, and identifying opportunities to better support the health workforce and strengthen health systems.

We studied the effects of environmental conditions on wellbeing of healthcare workers and quality of care, using a sample of rural clinics in Niger. Our specific objectives were to assess the effect of environmental conditions—or lack thereof—on (1) healthcare workers' wellbeing in terms of physical, mental, and social health and economic outcomes, and (2) quality of care.

## Methods

### Study design and conceptual framework

We conducted a descriptive case study in Niger using qualitative interviews with healthcare workers and other staff with job duties in the clinical setting (e.g., cleaners, waste handlers). We refer to any staff member who performed job duties in a clinical setting as a healthcare worker, as evidence in low-income countries suggests that individuals without formal medical training or job descriptions may assist in care delivery [12]. We considered the following to be environmental conditions: water, sanitation, hygiene, cleaning, waste management, energy (including lighting), and building design (e.g., floor plan, structural soundness, ventilation).

We examined the impacts of environmental conditions on healthcare workers' wellbeing as health and economic effects. We categorized health impacts into physical, mental, and social following the World Health Organization's (WHO) definition of health as "a state of complete physical, mental, and social well-being and not merely the absence of disease."[13] Table 1 provides definitions.

We examined the impacts of environmental conditions on quality of care using a framework from the Institute of Medicine [14]. The Institute of Medicine framework was initially proposed in a report that identified deficiencies in the quality of healthcare in the United States and identified six dimensions of quality care (safety, patient-centeredness, timeliness, efficiency, equity, and effectiveness), which could be targeted for improvement. These dimensions have since been widely applied in studies evaluating quality of care in the United States and other countries.

**Table 1. Categories of impacts of environmental conditions on healthcare workers' wellbeing.**

| Effects on wellbeing | Definition |
|---|---|
| Physical | Effects on the body from exposure to biological, chemical, or mechanical hazards |
| Mental | Effects on emotional or psychological wellbeing; primary effects are non-physical though physical sequelae may occur (e.g., chronic stress may impact cardiovascular health) |
| Social | Effects on relationships between healthcare workers and colleagues, patients, community members; effects on social standing or reputation |
| Economic | Effects on the income of healthcare workers, uncompensated labor, or other financial effects |

In this study, we excluded effectiveness, as it relates to treatment outcomes of specific medical procedures and is unlikely to be affected by environmental conditions. We define the remaining five dimensions in Table 2.

## Setting and study population

This study was conducted in partnership with the non-governmental organization World Vision. We purposively selected all HCFs in which World Vision had planned or executed improvements to environmental conditions as of October 2022, as part of an ongoing infrastructure improvement program. However, this study was not designed to be a program evaluation. Our sample included facilities at various stages of implementation, from no program components received, to partial implementation, to full program implementation. We asked healthcare workers to describe the effects of any environmental conditions that meaningfully impacted their wellbeing or quality of care, regardless of whether those conditions were affected by World Vision programming.

Selected HCFs were in the Dosso and Maradi regions of Niger. HCFs were small, rural clinics, serving a catchment area of approximately 5,000–40,000 people. All provided both outpatient and inpatient services, typically with 3–4 beds per HCF. Most employed 4–5 healthcare workers, and the facility director typically held a nursing degree.

As of October 2022, when data collection for this study was completed, surveys conducted by the research team (SI file 1) indicated that 64% of HCFs had an improved water source onsite, 28% had a water tower onsite, and 0% tested or treated water for microbial or chemical contamination. For sanitation, 92% had improved sanitation (ventilated improved pit latrines or pit latrines with slabs). Among facilities with latrines, 30% were cleaned daily, 39% were cleaned weekly, and 31% were cleaned less frequently or not at all. Only 40% reported that both water and soap were always available for washing, and 0% had hygienic hand drying materials. For waste management, 50% disposed sharps waste and 44% disposed infectious waste in a designated incinerator or burner, while the remainder disposed via open burning or dumping. For cleaning, 50% disinfected patient surfaces (e.g., beds) either daily or after every patient, with the remainder

**Table 2. Dimensions of impacts of environmental conditions on quality of care.**

| Quality of care measures | Definition |
|---|---|
| Safety | "Do no harm"; providing healthcare that maximizes benefits while minimizing harm |
| Patient-centeredness | Providing care that is respectful and responsive to patient needs and allows patient values to guide care |
| Timeliness | Reducing delays in care, particularly when those delays have potential to cause harm |
| Efficiency | Avoiding waste of time and resources (e.g., equipment, supplies) |
| Equity | Ensuring that quality of care is not affected by factors such as race, ethnicity, gender, religion, or personal characteristics |

disinfecting less frequently (weekly, monthly, or never); 44% sterilized reusable medical devices using only soap and water. Most facilities with environmental infrastructure had only gained access within the previous 6–12 months, typically through World Vision programming.

## Sampling

At each HCF, we asked the director to identify staff who worked in the following roles: (1) non-clinical roles for administration, supervision, supply chain management, and procurement related to environmental conditions; and (2) clinical roles interacting with patients and or performing other duties in clinical rooms (e.g., cleaning). From this list, we purposively selected and interviewed one person from each category, who completed one of two interview guides tailored to either clinical or non-clinical duties. For this study, we analyzed the subset of all individuals in clinical roles. All individuals who were approached to participate in the study agreed to participate.

## Data collection

We conducted interviews in two rounds in March and October 2022. Interview guides were similar across both rounds, with the second round incorporating additional probes and revising question wording to strengthen the richness of responses. Questions asked about what was good and what was challenging about environmental conditions, and how challenges affected care delivery. Participants were asked to recall specific instances where infrastructure had broken down or supplies (e.g., soap, tools for cleaning) had run out, then to reflect on how this impacted their wellbeing, ability to perform their job duties, interactions with patients, and functioning of the HCF overall.

A team comprising one interviewer and one notetaker conducted each interview at a private location within the HCF (e.g., office or vacant care room). Interviews lasted approximately 45–60 minutes. Where participants gave permission (n = 78, 96%), interviews were audio-recorded. The team interviewed participants in French, Hausa, or Zarma and transcribed recordings directly into French. We translated into English for analysis.

## Analysis

We conducted thematic analysis. First, we developed deductive codes using the constructs in Tables 1 and 2. Three authors applied these codes to a purposively selected set of three interviews from different HCFs, then met to discuss revisions to deductive codes and develop inductive codes for health impacts or quality of care impacts that were not identified a priori. A single author then coded the entire dataset.

## Ethics

This study was reviewed by the Institutional Review Board of University of North Carolina at Chapel Hill and approved by the Nigerien Ministry of Health. We received permission from the HCF director before beginning any data collection and obtained written informed consent from all participants.

## Results

### Study sample

Our study sample included 81 healthcare workers in 26 HCFs (Table 3). On average, participants had worked in their current position for 5.5 years and had worked in the health sector in any position for 7.5 years. A majority of our sample were facility administrators (directors, deputy directors, and health management committee members), though most administrators were also clinicians (e.g., doctor, head nurse).

**Table 3. Demographic characteristics of study participants.**

| Demographic characteristic | Sample size N (%) |
|---|---|
| *Gender* | |
| Male | 54 (66.6%) |
| Female | 27 (33.3%) |
| *Job title* | |
| Facility director or deputy director | 35 (43.3%) |
| Nurse | 20 (24.7%) |
| Cleaner | 14 (17.3%) |
| Health management committee member | 11 (13.6%) |
| Environmental health officer | 1 (1.2%) |
| *Years in current position** | |
| 0-2 years | 32 (40.5%) |
| 3-5 years | 22 (27.8%) |
| 6-10 years | 13 (16.5%) |
| >10 years | 12 (15.1%) |

*Two participants did not provide data on years of work experience.

## Effects on wellbeing

Table 4 summarizes the effects of environmental conditions by physical, mental, social, and economic wellbeing. In most cases, we frame effects in terms of how poor environmental conditions affected wellbeing, as lack of access was the norm. For physical, mental, and social wellbeing impacts, we observed that lack of adequate environmental conditions had almost exclusively detrimental effects, and improving conditions alleviated these effects. For economic impacts, effects were mixed. Lack of water, hygiene, and cleaning had economic costs associated with uncompensated duties for water fetching and out-of-pocket purchase of supplies like soap and detergents. However, presence of sanitation infrastructure and strengthened cleaning protocols also had economic costs, as staff (particularly cleaners) were expected to perform extra duties to operate and maintain new infrastructure without increases in salary.

## Physical

We identified three types of physical wellbeing impacts: biological, chemical, and mechanical. Biological impacts resulted from exposure to pathogens, such blood during deliveries or contamination on surfaces that were not properly disinfected. Chemical impacts resulted from exposure to toxic or corrosive chemicals. Mechanical impacts resulted from exposure to sharp or hot objects (i.e., cuts and burns) or musculoskeletal injury from lifting or carrying heavy objects. Physical impacts resulted primarily from exposure to hazards in one of these three categories. However, we also identified dehydration from lack of water as a physical health impact, which was not linked to direct biological, chemical, or mechanical hazards.

*Biological.* Biological impacts were most associated with poor hygiene and cleaning, though some participants were concerned about water-borne diseases. At nearly every HCF that lacked access to water, participants reported that they were unable to adequately perform hygiene and cleaning due to lack of water or energy to operate electrically powered pumps. Healthcare workers reported biological impacts for themselves but also for their families when they were unable to perform hygiene behaviors before leaving work:

**Table 4. Effects of environmental conditions on physical, mental, social, and economic wellbeing of healthcare workers.**

|  | Physical | Mental | Social | Economic |
|---|---|---|---|---|
| Water | Exposure to pathogens in unsafe drinking water<br>Injury and fatigue associated with water fetching<br>Dehydration from workdays without water | Dissatisfaction with working conditions | Arguments with patients around lack of water | Uncompensated labor for water fetching<br>Out-of-pocket costs to purchase water |
| Sanitation | Exposure to pathogens from open defecation and animal feces in the yard | Stress and worry about falling into collapsed latrine pits | Threats to dignity and reputation if observed while open defecating | ** Uncompensated labor for cleaning latrines |
| Hygiene | Exposure to pathogens in blood and other bodily fluids, particularly from working without PPE or inability to wash hands | Stress and worry about exposure to pathogens from working without PPE or exposing family members if unable to properly wash hands or bathe before returning home<br>Dissatisfaction with working conditions | Sense of self-worth or of being a "good" or "bad" healthcare worker because of ability to perform hand hygiene | Out-of-pocket costs for soap or PPE |
| Cleaning | Exposure to pathogens from dirty surfaces from lack of cleaning, or when performing cleaning duties without PPE | Stress and worry over spreading diseases to staff and patients through unclean surfaces<br>Dissatisfaction with working conditions | Reputational effects from dirty facility | Out-of-pocket costs for cleaning supplies<br>** Uncompensated labor for cleaning protocols |
| Waste management | Exposure to pathogens from needlesticks other sharps when handling waste<br>Inhalation of smoke from open burning<br>Cuts and burns from handling waste without PPE or necessary tools (e.g., wheelbarrows to transport waste) | Dissatisfaction with working conditions | Reputational effects from dirty facility | Uncompensated labor for digging waste pits |
| Energy | Operation of the water source to facilitate cleaning and hygiene | Dissatisfaction with working conditions | None reported | Out-of-pocket costs for utility bills |
| Building design | Exposure to airborne pathogens from overcrowding and poor ventilation<br>Injury (e.g., bites) and exposure to feces from animals that enter unfenced areas | Fear of physical or sexual assault in unsecured facilities, particularly among female healthcare workers at night<br>Dissatisfaction with working conditions | None reported | None reported |

Most effects were detrimental when environmental conditions were lacking and beneficial when strengthened. Asterisks indicate situations where improving environmental conditions had detrimental effects (**). PPE = personal protective equipment.

*What bothers me the most is that I went to the maternity ward to deliver a birth, and there was no water to wash my hands with soap or put on some [alcohol-based] gel. You see, and when I go back home, my children run to meet me, and I will touch them with my hands. . .. We don't know what microbes we put on the child. -Deputy director*

Less commonly, participants reported biological impacts associated with waste management, sanitation, and building design. Waste management exposed healthcare workers to pathogens when handling waste, as well as malaria vectors that bred in stagnant water accumulated in waste. Sanitation and building design were linked to exposure to fecal pathogens from open defecation and animals. In HCFs without a fence, livestock could roam the yard where patients waited and sometimes received treatment when interior rooms were overcrowded.

*Chemical.* Participants less frequently reported chemical health impacts compared to biological, and they were most associated with inadequate waste management and cleaning. Individuals responsible for waste management reported smoke inhalation associated with open burning. Some participants reported skin irritation (itching and burning) from improperly prepared cleaning solutions or contaminated water:

*You see the drinking water, it's a problem. You can wash yourself, but you'll really itch afterwards. We're really suffering. -Nurse*

*Mechanical*. Mechanical impacts were primarily associated with waste management (e.g., cuts or burns from transporting and burning waste). These impacts were most common when waste handlers lacked appropriate PPE or equipment to transport waste and resorted to carrying heavy loads on the head. In some instances, participants reported fatigue and injury from water fetching, particularly when the closest water source was several kilometers away and/or all water needed to be carried on foot:

*I was pregnant with my daughter. . .. I had to go to the school, where I told you I went to draw water. Every time I felt tired or sick, but I must fetch water.–Deputy director*

## Mental

We identified two types of mental impacts: satisfaction and stress. Satisfaction included healthcare workers' sense of enjoyment and contentment with the work environment. Stress included day-to-day worry, anxiety, or fears related to environmental conditions.

*Satisfaction*. Participants reported impacts on satisfaction for all environmental conditions, although satisfaction related to water was by far the most common and strongly felt. Participants reported feeling "dissatisfied," "unhappy," "uncomfortable," and "frustrated" about lack of water and described that conditions caused them to "suffer." Similarly, when HCFs gained access to water, participants reported feeling "very happy," "blessed," and "relieved." Water was a source of satisfaction because it enabled cleaning and hygiene activities, as well as basic medical tasks like delivering babies.

Healthcare workers were also dissatisfied when the working environment appeared dirty, untidy, and dysfunctional. This was most commonly related to poor cleaning, waste management, sanitation, and building design. Participants reported dissatisfaction with inadequate fencing that allowed wild animals, livestock, and unauthorized people to interrupt their work, and with cracked walls and leaking or collapsed roofs that created an unsafe and chaotic working environment:

*In the name of God how pitiful! Our rooms are in poor condition. Our rooms have a bad smell. Everywhere in the rooms when the rain falls. . . you are in the water trying to cross and bail the water out of the rooms. . . We suffer we suffer in the name of God. If you leave in the other room, you will notice even now the waste of the cows.–Cleaner*

Participants working at HCFs that had been renovated reported substantial increases in their satisfaction and pride in their working conditions.

*Stress*. Stress was most closely related to hygiene, cleaning, and lack of water to perform these activities. Typically, stress was caused by worry around unsafe working conditions, notably exposure to pathogens for working without adequate PPE, hygiene, or cleaning after procedures. Healthcare workers also reported stress around being unable to safely provide care or potentially harming patients or community members who visited the facility and were exposed to hazards in the building or yard (e.g., openly dumped sharps waste). Improving environmental conditions alleviated this stress.

Building design was also a source of stress in terms of security. Participants feared attacks from wild animals and assault from unauthorized persons who were able to enter unfenced areas. Women in particular feared assault when working alone at night. Many participants stressed the importance of a wall or fence for security and peace of mind.

*The doors don't have a lock to close. Even the windows, nothing is good. . .. Because there is a risk, our consciences are not at peace in the rooms that do not have a key. There are lunatics, and there are voiceless bandits. We are not happy like that. -Cleaner*

## Social

We identified one type of social impact related to reputation. Reputation impacts included healthcare workers' respect or social standing in the community because of environmental conditions.

*Reputation.* Reputational impacts were related to cleaning, hygiene, water, sanitation, and building design. These effects were most strongly influenced by visible cleanliness and tidiness of the building and yard, not necessarily surface contamination and sterilization that was not visible to the naked eye. Healthcare workers reported a sense of pride in HCFs that were visibly clean, well maintained, and stocked with water, PPE, and other essential supplies to deliver care:

*[Renovations to the HCF building] changed our experience positively. I work easily, because when I work, I know that I am in an ethical place. It's a source of pride. The environment is healthy. [Patients] come, and we treat with a clear conscience. -Director*

When facilities were not clean or well equipped, healthcare workers felt embarrassed to interact with patients and believed that their reputation would suffer. Waste management and sanitation were particularly relevant to controlling waste and feces in the yard, which were perceived to dimmish the respectability and reputation of the healthcare facility. More rarely, participants report verbal confrontations when asking patients or caregivers to supply their own water, soap, or PPE to compensate inadequate environmental conditions:

*There was blood there all night. [Patients] said hurtful words, saying that we are not doing our job, that we do nothing, that we are unable to stock even water. But that is not our fault. Even at home, I don't have water. -Head nurse*

## Economic

We identified two types of economic effects: out-of-pocket expenses and unpaid labor. Unpaid labor comprised tasks that healthcare workers did above and beyond their regular job duties without receiving additional salary. Out-of-pocket expenses comprised healthcare workers directly purchasing supplies for themselves or others.

*Unpaid labor.* Healthcare workers frequently reported performing extra unpaid duties outside their regular job descriptions to cope with inadequate environmental conditions. Unpaid labor was most commonly associated with lack of water, where participants reported traveling as far as ten kilometers round trip to fetch water, sometimes multiple times per day. When environmental conditions for cleaning and sanitation were improved, unpaid labor typically increased, as healthcare workers took on additional duties for cleaning and maintaining this infrastructure but did not receive increases in salary.

*Out-of-pocket expenses.* Out-of-pocket expenses were most associated with cleaning and hygiene. HCFs received government-issued allotments of hand soap, alcohol-based hand rub, and cleaning supplies, but these allotments were rarely sufficient. Nongovernmental organizations sometimes topped up allotments, but supplies often still ran low. To avoid stock outs,

healthcare workers paid out-of-pocket for soap, bleach, and other cleaning supplies that were available and affordable in local markets. Typically, costs were paid by senior healthcare workers (e.g., facility director, deputy director, or head nurse), who distributed supplies to more junior colleagues:

*There isn't [any fundings for cleaning supplies]. . .. If it's used up, we only tell him that there is no such thing, and he puts his hand in his pocket to give us what he has. -Cleaner*

Most HCFs had solar power, but in those paying an electricity utility bill, this was also commonly paid out-of-pocket by senior healthcare workers.

## Effects on job performance

We analyzed the effects of environmental conditions on quality of care in terms of safety, patient-centeredness, timeliness, efficiency, and equity. As with effects on wellbeing, we found that lack of environmental conditions reduced quality of care, and better environmental conditions improved it. Table 5 provides a summary.

*Safety.*The most common safety concern was exposure to pathogens due to inadequate cleaning and hygiene. However, participants described water as the most important environmental condition, because of its role in enabling all activities within the HCF:

*"We say that 'water is life,' because without water we can't do anything." -Health management committee president*

Waste management, sanitation, and building design were also cited as threats to safety. Inadequate waste management exposed visitors to burn pits, open dumps, or unsecured storage areas. Sanitation exposed visitors to fecal pathogens from humans and animals. Building design and overcrowding exposed individuals to pathogens from direct contact (e.g., patients forced to share beds) or inadequate ventilation. More rarely, cleaning was a concern for exposure to toxic or corrosive chemicals. In one instance, a healthcare worker described that visitors were non-lethally poisoned by improperly labeled cleaning solutions:

*Before we had pipes to draw from the tap, they [cleaners] take buckets of water and put bleach or hypochlorite water in it. We warn them [the villages] not to drink. . . but there are some who drink. . .. Sometimes there is even an overdose of bleach. -Nurse*

## Patient-centeredness

Patient-centeredness was most linked to cleaning. Healthcare workers reported that patient satisfaction was strongly influenced by visible cleanliness and tidiness, which was achieved through adequate cleaning (surface decontamination but not necessarily sterilization), waste management, sanitation (odors and visible soiling), and water for cleaning.

Patient-centeredness was also linked to privacy, dignity, and comfort due to water, sanitation, hygiene, and building design. Participants mentioned that without water, patients were thirsty and uncomfortable, particularly women in labor. Healthcare workers described the importance of ensuring that patients had privacy while defecating, bathing, and receiving certain medical procedures. Privacy for women during childbirth was especially prioritized and was facilitated by building designs with adequate numbers of treatment rooms, plus walls and floorplans that shielded women from observation or intrusion by outsiders:

**Table 5. Effects on quality of care.**

|  | Safety | Patient-centeredness | Timeliness | Efficiency | Equity |
|---|---|---|---|---|---|
| Water | Ability to perform IPC and cleaning protocols to reduce pathogen exposure | Patient satisfaction with water availability | Delays in care waiting for patients, caregivers, or healthcare workers to fetch water | Time to fetch water in facilities without an onsite source | Potential financial hardship on poor patients asked to purchase their own water |
| Sanitation | Exposure to fecal pathogens (human and animal) in the yard | Privacy of toilets Patient satisfaction with toilet cleanliness and odor | None reported | Time to travel to a private spot for open defecation | None reported |
| Hygiene | Exposure to pathogens related to adherence to handwashing and IPC protocols | Privacy of bathing facilities, particularly for pregnant women after giving birth | Delays in care waiting for patients, caregivers, or healthcare workers to procure soap or PPE | Time to procure supplies (hand soap, alcohol-based hand rub, PPE) in situations of low or no stock | Potential financial hardship on poor patients asked to purchase their own soap or PPE |
| Cleaning | Exposure to pathogens related to adherence to cleaning protocols Injuries from contact with or consumption of improperly labeled cleaning solutions | Patient satisfaction with cleanliness (visible soiling and smells) | None reported | Time to to procure supplies (detergents, bleach) in situations of low or no stock | Potential financial hardship on poor patients asked to purchase their own cleaning supplies |
| Waste management | Exposure to pathogens on infectious waste Injuries from sharps waste or burn pits | Patient satisfaction with tidiness and waste accumulation in the yard | None reported | Time to dig pits for open burning | None reported |
| Energy | Ability to perform IPC and cleaning protocols dependent on electrically-pumped water | None reported | None reported | Ability to operate infrastructure dependent on electricity Sufficiency of lighting to perform medical procedures | None reported |
| Building design | Overcrowding and exposure to airborne pathogens | Satisfaction related to overcrowding and privacy of rooms | Ability to appropriately triage patients due to lack of rooms | Damage or spoiling of medical equipment and supplies from collapsed or leaking roofs, windows, and walls | None reported |

Most effects were detrimental when environmental conditions were lacking and beneficial when strengthened. Asterisks indicate situations where improving environmental conditions had detrimental effects (**). PPE = personal protective equipment, IPC = infection prevention and control.

*For example, here, there is a delivery room next to the consultation room. That's not good at all. . . . What is this failure? It's not suitable because you can see everything the woman does in the delivery room. -Cleaner*

## Timeliness

Healthcare workers reported delays in care associated with lack of water or hygiene (most commonly hand soap or gloves). When supplies were lacking, healthcare workers sent patients or their caregivers to fetch water or purchase soap at local markets, or healthcare workers would write patients a prescription for gloves to be filled at the pharmacy before providing treatment. Delays in care were also reported when healthcare workers left the facility in search of these supplies themselves, and a patient arrived while the facility was unattended:

*Before we had to go out for our [water] needs. As a result, the patients also had to wait a long time before our return, because the distance is long. But now it's very close; as soon as you hear someone's voice, you can quickly leave. -Facility director*

In emergency situations where treatment was urgently needed, most healthcare workers simply provided care without water or PPE. Healthcare workers universally reported that the

HCF would remain open regardless of the environmental conditions, and they would simply make do with the resources available.

## Efficiency

Healthcare workers reported effects on efficiency related to all environmental conditions. The greatest impact on efficiency was water fetching due to lack of a functioning onsite water source. Participants reported multiple trips per day to fetch water—either themselves or by others. Healthcare workers made trips to nearby villages to purchase supplies for hygiene and cleaning when normal supply chains failed to provide adequate stock. In HCFs without sanitation, healthcare workers would walk to open defecation sites. Substantial effort was dedicated to digging pits for open burning of waste, though this was typically done by paid or volunteer laborers, not healthcare workers themselves.

Energy was critical for ensuring that systems dependent on electricity remained operational, such as electrical pumping for water. Multiple participants reported that substantial investments had been made in water towers that were non-functional because electrical pumping systems had broken down. Energy for lighting was important for providing care at night, especially for deliveries and emergency care:

*In the past, we give birth with a flashlight by putting it in the mouth to light up, to assist in childbirth. Now, thank God, everywhere there is light, we work with peace of mind. -Nurse*

Building design had substantial effects on efficiency. Structural failures in walls and roofs left medical equipment, supplies, and records exposed to wind, rain, sand, and other debris that caused damage and made entire rooms unusable. More broadly, dissatisfaction with environmental conditions contributed to inefficiency at the health system level, as healthcare workers reported challenges attracting and retaining staff:

*We're really suffering. We're only staying because they haven't had anyone Who's going to stay here. That's why they left us here for 8 years. In the whole district, there is not someone who has lasted like us.–Deputy director*

## Equity

We found few examples of healthcare workers reporting that environmental conditions directly affected equity. Healthcare workers did report asking patients and caregivers to bring or purchase supplies for their own care—most commonly water, hand soap, gloves, and cleaning supplies—which is a potential barrier to care for poorer households.

## Discussion

We assessed the effects of environmental conditions on healthcare workers' wellbeing and quality of care, using qualitative interviews with 81 healthcare workers at 26 rural HCFs in Niger. The norm in HCFs of our sample was poor environmental conditions, and healthcare workers primarily reported negative effects on wellbeing and quality of care. In HCFs where environmental conditions improved, participants reported corresponding improvements in wellbeing and quality of care in most cases.

### Effects on healthcare worker wellbeing

We examined effects on healthcare workers in terms of health (physical, mental, and social) and economic impacts. Participants consistently reported that poor environmental conditions

decreased physical wellbeing. Exposure to pathogens was associated with all environmental conditions, though hygiene and cleaning were of greatest concern to participants. These results are unsurprising, given a wealth of literature linking poor environmental conditions to exposure and infection rates, particularly for hand hygiene and cleaning [15–17]. Current WHO guidelines include environmental conditions as a core component of infection prevention and control programs, with emphasis on hygiene, cleaning, water, sanitation, and waste management [18]. Our findings also highlight the critical role that water, energy, and building design play in facilitating cleaning and hygiene.

We also documented impacts on physical wellbeing from chemical and mechanical hazards, which corroborates prior studies [19–22]. Guidelines for waste management are designed to mitigate risk of needlesticks, cuts, and other injuries and the dangers of fumes generated during incineration [23]. In Niger, we found that these risks were exacerbated due to recombining waste streams after segregation for open burning and lack of adequate PPE or equipment to transport waste (e.g., wheelbarrows). We also found water fetching to be a source of musculoskeletal injury. Previous studies have documented injury during household water fetching [24]. Increased quantities of water needed in HCFs compared to households likely increase the risk of injury, either because loads carried are larger or trips are more frequent.

This study contributes several novel findings about the effects of environmental conditions on mental and social wellbeing. We found that all environmental conditions affected healthcare worker satisfaction, and in some cases contributed to deeper levels of chronic stress and fatigue. To our knowledge, this study is the first to document links between healthcare worker stress and environmental conditions, though one previous study has shown dissatisfaction with poor environmental conditions [9].

Prior studies have suggested that healthcare workers suffering from stress, fatigue, and burnout from other factors (e.g., workload, understaffing, bullying, unrealistic patient expectations) are less likely to deliver quality care, and that improving the underlying causes of these conditions can improve outcomes like medical errors and healthcare-acquired infections [25–27]. These are serious concerns for the health and wellbeing of both healthcare workers and their patients, which warrant further study and development of effective interventions.

Economic impacts were the only impacts where improving environmental conditions was both beneficial and detrimental. Improving environmental conditions for water, cleaning, hygiene, and waste management improved economic wellbeing by reducing out-of-pocket expenses and unpaid labor for water fetching and digging and managing open burning pits. However, improving environmental conditions for sanitation and cleaning reduced economic wellbeing by imposing additional duties on healthcare workers to operate and maintain these systems without increased wages. We did not attempt to value the net effect, but we suspect that overall improvements to environmental conditions enhance economic wellbeing for healthcare workers. Daily water fetching was time consuming and completely eliminated by an onsite water source, while cleaning was already part of regular duties and increased only marginally.

## Effects on quality of care

We examined effects on quality of care in terms of safety, patient-centeredness, timeliness, efficiency, and equity [14]. Safety correlated strongly with impacts on healthcare workers' wellbeing. The same biological, chemical, and mechanical hazards that affected healthcare workers' physical wellbeing impacted safety of care in similar ways. These findings are in line with guidelines on patient safety and quality of care that note the importance of environmental conditions [28,29].

Our work highlights the importance of effects beyond safety and reveals several novel effects. Environmental conditions influenced patient centeredness of care. Healthcare workers reported reputation as a social wellbeing impact, and described how facilities that were dirty, untidy, or lacking water and other key supplies were perceived to provide low-quality service and occasionally led to confrontations with patients and caregivers. We did not triangulate these findings with patient interviews, though previous studies have suggested links between water, sanitation, and hygiene and patient satisfaction and care seeking behavior [7]. Our results add support to the theory that environmental conditions are important for patient centeredness, satisfaction, and health seeking.

Timeliness is also key for quality of care but has been rarely considered in previous studies of environmental conditions in HCFs. We found instances of reduced timeliness where healthcare workers had left to fetch water, travel to open defecation sites, or do other tasks to compensate for inadequate environmental conditions. However, these delays were transient and infrequent. Generally, healthcare workers in our study reported that facilities always remained open, regardless of infrastructure breakdowns or stock outs. This represents an important trade-off between timeliness and safety. This trade-off was common in Niger for all procedures, regardless of their level of urgency. However, in some situations, delaying non-emergency procedures until breakdowns or stockouts are resolved may be lower risk. Research to quantify how environmental conditions contribute to different quality of care outcomes could help develop risk models and inform clinical decision making in low-resource contexts.

We found substantial impacts of environmental conditions on efficiency. Our study was not designed to quantify these effects, but we anticipate their economic value is substantial. Billions of dollars in economic value are lost annually from time spent walking to and waiting at communal water sources [30]. These figures are estimated just for household water collection and calculated on market wages for casual labor. The value of time savings for environmental conditions in HCFs is likely greater, given higher wages of healthcare workers and additional tasks for fetching other supplies for cleaning and hygiene. Rehabilitating HCF buildings to fix roof leaks and damaged walls may plausibly be cost-saving, depending on the value of damages to medical equipment and supplies relative to the costs of rehabilitation, though further research is needed to more precisely value costs and benefits.

We observed impacts on quality of care at the HCF level, but environmental conditions have implications for broader health systems security and resilience. Inadequate conditions undermine the ability of HCFs to rapidly and appropriately respond to population health needs. Healthcare workers in our sample coped with day-to-day challenges through adaptations like out-of-pocket purchase of supplies. On the surface, presence of supplies like soap and PPE may suggest that environmental conditions are adequate. However, these conditions are currently achieved through shifting the responsibility from health systems to individual healthcare workers. These solutions are neither sustainable nor scalable, particularly in response to growing threats from climate change and epidemics and the need for resilience at the health-systems level [31]. The government of Niger has recently released a national WaSH in HCFs strategy, which recognizes the need to include environmental conditions as part of health systems strengthening though specific funding sources and policy implementation mechanisms have not yet been determined [32].

## Limitations

This study evaluated healthcare workers in the context of an ongoing program delivered by a non-governmental organization. Participants may have been influenced by social-desirability bias to exaggerate the effects of the program or to exaggerate challenges in hopes of receiving

additional programming. We did take steps to mitigate possible bias. The data collection team was an independent consulting firm not affiliated with the implementing organization. Enumerators emphasized their role as impartial evaluators and informed participants that responses would not affect current or future program eligibility.

Our analysis was designed to compile a framework of self-reported effects of environmental conditions. We did not assess frequency of these effects, nor did we attempt to verify these effects through other methods, such as triangulation with patients. We anticipate that the prevalence of different effects will vary by HCF type and level of access to environmental conditions, and further research is needed to assess causality and strength of relationships. Development of reliable quantitative measures will be needed to advance this research.

## Implications for research and practice

**Monitor beyond infectious outcomes.** We identified a variety of impacts of environmental conditions, including, but not limited to, pathogen exposure and healthcare-acquired infections. Prior research has heavily emphasized microbiological outcomes (e.g., surface contamination measured through contact plates or swabbing) and infection rates as the primary outcomes for impact evaluations [6]. However, these measures are expensive, difficult to capture in a single cross-sectional measurement due to high variability, and require large sample sizes to achieve adequate power [33].

Our research identifies several possible areas to develop impact measures that are low-cost and reliable in cross-sectional surveys. Indicators for stress and burnout are relevant for healthcare workers, and indicators for satisfaction and care seeking likely apply to patients. Importantly, we found that many impacts may not necessarily correlate with microbiological indicators of surface contamination or infection risk. Visible cleanliness (i.e., decontamination but not necessarily sterilization), tidiness, organization, and security of the work environment were all important influences on stress, satisfaction, and patient-centeredness of care that are likely independent or weakly correlated with microbial outcomes.

Furthermore, we found that environmental conditions have substantial effects on efficiency of care, independent of safety. Indicators such as time savings of healthcare workers, frequency of breakdowns, time to restore functionality, and other measures of efficiency are highly relevant to HCF and health systems functioning that would be valuable measures of program impact. They also offer an alternative to microbial measures that is low cost and feasible in low-resource settings. Research to develop a reliable set of indicators for non-infectious outcomes would support both research and practice.

**Re-evaluate cost-benefit.** This study revealed several effects of environmental conditions that have not been previously identified, particularly related to mental, social, and economic wellbeing and efficiency and timeliness of care. These findings suggest a need to reconsider how researchers, policy makers, and practitioners evaluate the costs and benefits of programs on environmental conditions in HCFs.

We draw parallels with prior research on benefits of water, sanitation, and hygiene (WaSH) programs in community settings. Impact evaluations of community WaSH that focused on a narrow set of infectious health impacts and related sequelae have shown little to no effect [34–36], leading to debate on the overall benefits of WaSH programs [37,38]. Yet cost-benefit evaluations that more holistically consider non-infectious health and economic impacts (e.g., time savings) conclude that WaSH is favorable from a cost-benefit perspective for human health and development [30].

We argue that the same is likely true for HCFs. Impact evaluation to-date has focused on a narrow set of outcomes for healthcare-acquired infections (and intermediate measures of

surface contamination) and health seeking. Yet evidence that environmental conditions reduce healthcare-acquired infections is tenuous—with the exception of hand hygiene, for which evidence is more robust [6,15]. Even completely absent any effect of healthcare-acquired infections, impacts on mental and social wellbeing of healthcare workers and patient-centeredness, timeliness, and efficiency are important for ensuring high quality care and a well-functioning health system.

Environmental conditions are often deprioritized in low-resource settings, compared to other expenses that have more conspicuous effects (e.g., purchase of drugs) [28]. High upfront costs of infrastructure construction can be a deterrent. Yet studies have suggested that the cost of basic access to water, sanitation, hygiene, and waste management is relatively modest compared to total expenditures in the health sector [39,40]. Pairing costs data with robust estimates of benefits can allow policy makers to prioritize investments with the greatest wellbeing impacts and create strong and resilient health systems. Research is needed to develop quantitative measures of the benefits of environmental conditions to inform these decisions.

## Conclusion

We evaluated the effects of environmental conditions on wellbeing and job performance of healthcare workers. We found that poor environmental conditions had detrimental effects on physical, mental, social, and economic wellbeing of healthcare workers, and improving conditions alleviated these effects. Similarly, poor environmental conditions had detrimental effects on quality of care in terms of safety, efficiency, timeliness, patient centeredness, and equity.

Effects on physical wellbeing and safety of care have been well documented and are recognized in international guidelines on infection prevention and safety of care [18,28,29,41]. However, our study is the first to rigorously document impacts beyond physical health and infection risk. Our research suggests that environmental conditions have substantial effects on mental and social wellbeing of healthcare workers and on efficiency, timeliness, and patient-centeredness of care. Inefficiency due to lost time and damaged supplies and equipment likely have substantial economic value and warrant further study to reconsider cost-benefit calculations. We recommend that future research and decision making for policy and practice incorporate more holistic indicators to evaluate the broad range of effects of environmental conditions.

## Supporting information

**S1 Data. Facility demographic data.**
(CSV)

## Acknowledgments

We thank Kaida Liang, Ben Tidwell, Seth Marcus, and Peter Hynes for their support in funding acquisition, program management, and logistics of field data collection. We thank Irving Hoffman for his feedback on early versions of the data collection tools. We thank the enumerators who conducted the interviews and staff of the CERISES team for provided support and guidance on field logistics. We thank the healthcare workers in Niger who generously provided their time to participate in this study.

## Author Contributions

**Conceptualization:** Darcy M. Anderson, Aaron Salzberg.

**Data curation:** Valerie Bauza.

**Formal analysis:** Darcy M. Anderson.

**Funding acquisition:** Aaron Salzberg.

**Investigation:** Darcy M. Anderson.

**Methodology:** Darcy M. Anderson, Valerie Bauza, Lucy Tantum.

**Project administration:** Darcy M. Anderson, Ezechiel Mahamane, Valerie Bauza, Kairou Oudou Bilo Mahamadou.

**Supervision:** Ezechiel Mahamane, Kairou Oudou Bilo Mahamadou.

**Writing – original draft:** Darcy M. Anderson.

**Writing – review & editing:** Darcy M. Anderson, Ezechiel Mahamane, Valerie Bauza, Kairou Oudou Bilo Mahamadou, Lucy Tantum, Aaron Salzberg.

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
