## [Decision Letter · Decision Letter 0]

28 Sep 2023

PGPH-D-23-01405

Effects of environmental conditions on healthcare worker wellbeing and quality of care: a qualitative study

Dear Dr. Anderson,

Thank you for submitting your manuscript to PLOS Global Public Health. After careful consideration, we feel that it has merit but does not fully meet PLOS Global Public Health’s publication criteria as it currently stands. Therefore, we invite you to submit a revised version of the manuscript that addresses the points raised during the review process.

Please address all reviewer comments located at the bottom of this email.

We look forward to receiving your revised manuscript.

Kind regards,

Natalia M. Rodriguez, PhD, MPH

Academic Editor

Journal Requirements:

1. Please provide additional details regarding participant consent. In the ethics statement in the Methods and online submission information, please ensure that you have specified (1) whether consent was informed and (2) what type you obtained (for instance, written or verbal, and if verbal, how it was documented and witnessed). If your study included minors, state whether you obtained consent from parents or guardians. If the need for consent was waived by the ethics committee, please include this information.

2. Since your data is not available for proprietary reasons, please explain via email why the data is not available. Please also include the contact information for the third party organization that should be contacted should other researchers want to request access to this data and please include the full citation of where the data can be found. We also request that you verify with us via email that any researcher will be able to obtain the data set in the same manner that the you have obtained it. If you feel you are unwilling or unable to adhere to this policy, please explain your reasons by return email and your exemption request will be escalated to the editor for approval. Your exemption request will be handled independently and will not hold up the peer review process, but will need to be resolved should your manuscript be accepted for publication. One of the Editorial team will be in touch if they require more information.

Additional Editor Comments (if provided):

Reviewers' comments:

Reviewer's Responses to Questions

**Comments to the Author**

1. Does this manuscript meet PLOS Global Public Health’s publication criteria? Is the manuscript technically sound, and do the data support the conclusions? The manuscript must describe methodologically and ethically rigorous research with conclusions that are appropriately drawn based on the data presented.

Reviewer #1: Partly

Reviewer #2: Partly

2. Has the statistical analysis been performed appropriately and rigorously?

Reviewer #1: No

Reviewer #2: Yes

3. Have the authors made all data underlying the findings in their manuscript fully available (please refer to the Data Availability Statement at the start of the manuscript PDF file)?

Reviewer #1: No

Reviewer #2: No

4. Is the manuscript presented in an intelligible fashion and written in standard English?

Reviewer #1: Yes

Reviewer #2: Yes

5. Review Comments to the Author

Reviewer #1: Summary: This is a very important issue in health care environment for patient’s service and management especially poor environmental conditions. I appreciate to authors to convey this essential issue for health sectors. However study is known phenomena in poor environmental conditions had detrimental effects on quality of care in terms of safety, efficiency, timeliness, patient centeredness, and equity.

Major Issues: Type of case study may be mention in the method section.

Minor Issues: Data analysis and presentation may place in different way.

Other Comments: Curious to know both interviews had taken from same person and questionnaire.

Reviewer #2: The study is much relevant in the context of resource limited countries. It will be open warming i context of securing the health and safety of both health workers, support staffs, the patients and the attendants in the health care settings.

There are few suggestions and comments to improve the manuscript though it reflects hard work of the study team.

Authors are strongly requested to include line numbers in the manuscript text before submission in future or after revision of this manuscript.

DMA is supported by a grant from the National Institutes of Environmental Health (T32ES007018)

-Disclose if his funder have any role in decision to publish the work?

Regarding data availability statement, please adhere to Journal guidelines. It is true that

participants identity must not be revealed in any case. But coded data could be made available on request by

ensuring the confidentiality of participants identity.

Please specify if the first line in the introduction section have specific reference i.e. for the components of

environmental condition in HCFs.

Methods well described. But cleaners and support staffs cannot be called healthcare workers.

Please describe in brief the framework from the Institute of Medicine,[13] (-you can include in the annex the detail but give a brief in the

main text)

Please include appropriate references in study site description. Are the data described obtained from World Vision

program or from National data?

In Sampling please indicate how the sampling was done (i.e. how much HCFs were selected and the number of interviews or observations were conducted).

You could also want to include response rate.

Results section;

Results well formatted and well summarized in the tables but should be backed up by rich verbatim n the descriptions.

Please include reliable verbatim on the physical impact on wellbeing.

Please include relatable verbatim as well in Mental impact (Stress). The finding will be much relatable and strong by the use of appropriate verbatim and readers

can visualize the scenario which is the worth of qualitative studies.

If applicable please include relatable pictures in the reliable sections or include them in the annexes (please adhere to Journal criteria while adding pictures and graphics)

In the introduction of discussion section; the last line would be The facilities where environmental conditions were improved, participants also reported corresponding improvements in wellbeing and quality of care in most cases.

Limitations well explained

Implications well described with strong recommendations for future research.

6. PLOS authors have the option to publish the peer review history of their article (what does this mean?). If published, this will include your full peer review and any attached files.

**Do you want your identity to be public for this peer review?** For information about this choice, including consent withdrawal, please see our Privacy Policy.

Reviewer #1: **Yes: **A K M Rabiul Hasan

Reviewer #2: **Yes: **Rabindra Bhandari

---

## [Editor Report · Decision Letter 1]

22 Nov 2023

Effects of environmental conditions on healthcare worker wellbeing and quality of care: a qualitative study in Niger

PGPH-D-23-01405R1

Dear Ms. Anderson,

We are pleased to inform you that your manuscript 'Effects of environmental conditions on healthcare worker wellbeing and quality of care: a qualitative study in Niger' has been provisionally accepted for publication in PLOS Global Public Health.

Best regards,

Natalia M. Rodriguez, PhD, MPH

Academic Editor